# Diagnostic accuracy of cervical cancer screening and screening–triage strategies among women living with HIV-1 in Burkina Faso and South Africa: A cohort study

Helen A. Kelly[1,2]*, Admire Chikandiwa[3], Bernard Sawadogo[4], Clare Gilham[2], Pamela Michelow[5,6], Olga Goumbri Lompo[4], Tanvier Omar[5,6], Souleymane Zan[7], Precious Magooa[8], Michel Segondy[9], Nicolas Nagot[9], Nicolas Meda[4], Sinead Delany-Moretlwe[3], Philippe Mayaud[2], for the HARP Study Group[¶]

1 Catalan Institute of Oncology, Bellvitge Biomedical Research Institute (IDIBELL), L'Hospitalet de Llobregat, Barcelona, Spain, 2 London School of Hygiene & Tropical Medicine, London, United Kingdom, 3 Wits Reproductive Health and HIV Institute, Faculty of Health Sciences, University of the Witwatersrand, Johannesburg, South Africa, 4 Centre de Recherche Internationale en Santé, University of Ouagadougou, Burkina Faso, 5 Department of Anatomical Pathology, University of the Witwatersrand, Johannesburg, South Africa, 6 National Health Laboratory Service, Johannesburg, South Africa, 7 Department of Gynaecology, Centre Hospitalier Universitaire Yalgado, Ouagadougou, Burkina Faso, 8 Sexually Transmitted Infections Reference Centre, National Institute for Communicable Diseases, National Health Laboratory Service, Johannesburg, South Africa, 9 UMR1058, Montpellier University, Montpellier, France

¶ Membership of the HARP Study Group is provided in the Acknowledgements.
* helen.kelly@lshtm.ac.uk

**Data Availability Statement:** All data files will be available on the Mendeley online repository at

## Abstract

### Background

Cervical cancer screening strategies using visual inspection or cytology may have suboptimal diagnostic accuracy for detection of precancer in women living with HIV (WLHIV). The optimal screen and screen–triage strategy, age to initiate, and frequency of screening for WLHIV remain unclear. This study evaluated the sensitivity, specificity, and positive predictive value of different cervical cancer strategies in WLHIV in Africa.

### Methods and findings

WLHIV aged 25–50 years attending HIV treatment centres in Burkina Faso (BF) and South Africa (SA) from 5 December 2011 to 30 October 2012 were enrolled in a prospective evaluation study of visual inspection using acetic acid (VIA) or visual inspection using Lugol's iodine (VILI), high-risk human papillomavirus DNA test (Hybrid Capture 2 [HC2] or careHPV), and cytology for histology-verified high-grade cervical intraepithelial neoplasia (CIN2+/CIN3+) at baseline and endline, a median 16 months later. Among 1,238 women (BF: 615; SA: 623), median age was 36 and 34 years ($p < 0.001$), 28.6% and 49.6% ever had prior cervical cancer screening ($p < 0.001$), and 69.9% and 64.2% were taking ART at enrolment ($p = 0.045$) in BF and SA, respectively. CIN2+ prevalence was 5.8% and 22.4% in BF and SA ($p < 0.001$), respectively. VIA had low sensitivity for CIN2+ (44.7%, 95% confidence interval [CI] 36.9%–52.7%) and CIN3+ (56.1%, 95% CI 43.3%–68.3%) in both

doi:10.17632/yd5ygw38vj.1 (under embargo until 1 Jan 2021).

**Funding:** The research leading to these results has received funding from the European Commission (EC) 7th Framework Programme under grant agreement No. HEALTH-2010-F2-265396 (https://ec.europa.eu/research/fp7) awarded to PM. The funder did not contribute in the study design, data collection and analysis, decision to publish, or preparation of the manuscript.

**Competing interests:** The authors have declared that no competing interests exist.

**Abbreviations:** ASCUS+, atypical squamous cells of undetermined significance, or greater; BF, Burkina Faso; CHU-Yalgado, Centre Hospitalier Universitaire Yalgado; CIN, cervical intraepithelial neoplasia; HC2, Hybrid Capture 2; HPV, human papillomavirus; HR, high-risk; HR-HPV, high-risk human papillomavirus; HSIL+, high-grade squamous intraepithelial lesion or greater; ICC, invasive cervical cancer; LMICs, low- and middle-income countries; LSIL+, low-grade squamous intraepithelial lesion or greater; NNR, number needed to refer; NPV, negative predictive value; PPV, positive predictive value; RLU, relative light unit; RSen, relative sensitivity; RSpec, relative specificity; SA, South Africa; STI, sexually transmitted infection; VIA, visual inspection using acetic acid; VILI, visual inspection using Lugol's iodine; WLHIV, women living with HIV.

countries, with specificity for ≤CIN1 of 78.7% (95% CI 76.0%–81.3%). HC2 had sensitivity of 88.8% (95% CI 82.9%–93.2%) for CIN2+ and 86.4% (95% CI 75.7%–93.6%) for CIN3+. Specificity for ≤CIN1 was 55.4% (95% CI 52.2%–58.6%), and screen positivity was 51.3%. Specificity was higher with a restricted genotype (HPV16/18/31/33/35/45/52/58) approach (73.5%, 95% CI 70.6%–76.2%), with lower screen positivity (33.7%), although there was lower sensitivity for CIN3+ (77.3%, 95% CI 65.3%–86.7%). In BF, HC2 was more sensitive for CIN2+/CIN3+ compared to VIA/VILI (relative sensitivity for CIN2+ = 1.72, 95% CI 1.28–2.32; CIN3+: 1.18, 95% CI 0.94–1.49). Triage of HC2-positive women with VIA/VILI reduced the number of colposcopy referrals, but with loss in sensitivity for CIN2+ (58.1%) but not for CIN3+ (84.6%). In SA, cytology high-grade squamous intraepithelial lesion or greater (HSIL +) had best combination of sensitivity (CIN2+: 70.1%, 95% CI 61.3%–77.9%; CIN3+: 80.8%, 95% CI 67.5%–90.4%) and specificity (81.6%, 95% CI 77.6%–85.1%). HC2 had similar sensitivity for CIN3+ (83.0%, 95% CI 70.2%–91.9%) but lower specificity compared to HSIL+ (42.7%, 95% CI 38.4%–47.1%; relative specificity = 0.57, 95% CI 0.52–0.63), resulting in almost twice as many referrals. Compared to HC2, triage of HC2-positive women with HSIL+ resulted in a 40% reduction in colposcopy referrals but was associated with some loss in sensitivity. CIN2+ incidence over a median 16 months was highest among VIA baseline screen-negative women (2.2%, 95% CI 1.3%–3.7%) and women who were baseline double-negative with HC2 and VIA (2.1%, 95% CI 1.3%–3.5%) and lowest among HC2 baseline screen-negative women (0.5%, 95% CI 0.1%–1.8%). Limitations of our study are that WLHIV included in the study may not reflect a contemporary cohort of WLHIV initiating ART in the universal ART era and that we did not evaluate HPV tests available in study settings today.

## Conclusions

In this cohort study among WLHIV in Africa, a human papillomavirus (HPV) test targeting 14 high-risk (HR) types had higher sensitivity to detect CIN2+ compared to visual inspection but had low specificity, although a restricted genotype approach targeting 8 HR types decreased the number of unnecessary colposcopy referrals. Cytology HSIL+ had optimal performance for CIN2+/CIN3+ detection in SA. Triage of HPV-positive women with HSIL+ maintained high specificity but with some loss in sensitivity compared to HC2 alone.

## Author summary

### Why was this study done?

- Invasive cervical cancer is the second most common cancer among women in low- and middle-income countries and a leading cause of cancer-related death in women in sub-Saharan Africa.

- Women living with human immunodeficiency virus (WLHIV) have an increased risk of cervical cancer and precancer. The majority of WLHIV live in low- and middle-income countries, where access to cervical cancer screening and treatment of precancerous cervical lesions is limited.

- Screening approaches most commonly used in sub-Saharan Africa, including visual inspection of the cervix and cervical cytology, which checks for cervical cell abnormalities, have shown variable diagnostic accuracy to detect precancerous lesions. Molecular-based screening approaches, such as human papillomavirus (HPV) DNA testing, which screens for oncogenic HPV infection, have shown high sensitivity for cervical precancer but can result in over-referral to colposcopy, a procedure to determine eligibility for treatment.

- The optimal screening test, age to begin screening, and frequency of screening for WLHIV remain uncertain.

## What did the researchers do and find?

- We evaluated several cervical cancer screening strategies in over 1,200 WLHIV in sub-Saharan Africa.

- We found that an HPV DNA test identified a greater number of women with precancer compared to visual inspection and cytology methods. However, there was a greater proportion of women without precancer who had a positive HPV DNA test, meaning a triage test using cytology or visual inspection was required to determine treatment eligibility.

- Simple user-applied modifications to the HPV-DNA-based test resulted in fewer women without precancer testing positive and fewer women needing triage.

- In settings with adequate infrastructure, cervical cytology was a useful triage test for HPV-positive women.

## What do these findings mean?

- Our data contribute to the evidence on choice of screening strategies for detection of cervical precancer among WLHIV in low- and middle-income settings.

- HPV DNA tests can play an important role in cervical cancer screening, especially in the era of universal antiretroviral therapy and where availability of self-sampling will facilitate screening participation.

- Prevention of cervical cancer should rank high as a public health priority in sub-Saharan Africa since WLHIV represent a group at very high risk of cervical precancer.

## Introduction

In May 2018, the Director-General of the World Health Organization (WHO) announced a global call for action towards the elimination of invasive cervical cancer (ICC) as a public health problem, calling for more innovative technologies for detection of high-grade cervical intraepithelial neoplasia (CIN) grade 2 and higher (CIN2+) and better strategies to increase screening coverage and uptake [1]. The 2030 cervical cancer elimination targets include vaccinating 90% of eligible girls against human papillomavirus (HPV), screening 70% of eligible women for cervical cancer, and effectively treating 90% of those with a positive lesion [2].

Since the introduction of HPV vaccination, cervical cancer screening in high-income settings has shifted from the identification of cellular changes in cytology towards the molecular detection of high-risk HPV (HR-HPV) types as the form of primary screening, allowing for increased automation and standardisation of diagnostic procedures. Studies among regularly screened women in Europe have shown that HPV-based screening reduces the risk of ICC by up to 70% compared to cervical cytology, also allowing extension of screening intervals due to the higher negative predictive value (NPV) [3]. Approaches using HPV DNA tests can be easily adapted to resource-limited settings, allow for self-collected samples, and are less observer dependent than visual inspection methods, which have variable sensitivity and specificity among women living with HIV (WLHIV) [4–6]. However, HPV DNA tests can detect many transient infections, meaning that their specificity for high-grade CIN is low, especially in populations with high prevalence of HR-HPV [7]. This is problematic among WLHIV, who are more likely to have multiple HR-HPV co-infections with a broader range of HR-HPV genotypes [8] and have a higher risk of HR-HPV incidence and persistence compared to HIV-negative women [9]. However, there is increasing evidence that WLHIV on effective ART with sustained HIV viral suppression have lower prevalence of HR-HPV [10], which may impact on diagnostic accuracy of HPV-DNA-based testing in screening and screening–triage for CIN2+ detection. Current WHO guidelines recommend that cervical cancer screening should be started in sexually active girls and women, as soon as they have tested positive for HIV, and if the screening test is negative, a repeat test should be done within 3 years [11], although more recent evidence from cross-sectional and prospective studies is being considered in the revision of these recommendations, including optimal screen and screen–triage modalities, age to initiate screening, and screening intervals.

The majority of WLHIV live in low- and middle-income countries (LMICs), where cervical cancer incidence is high [12] but where cervical cancer screening coverage and linkage to care is low [13,14] and largely unknown for WLHIV [15], as infrastructure and personnel requirements for screening and management put high demands on the health systems. Furthermore, cervical cancer screening approaches more commonly utilised in LMICs, including visual inspection methods and cervical cytology, have variable and often suboptimal sensitivity and specificity for CIN2+ detection and can lack reproducibility both in women in the general population and among WLHIV. Screening and screening–triage strategies that can be feasibly implemented and that have high diagnostic accuracy to detect CIN2+/CIN3+ are needed. We previously evaluated the association of HIV-related factors with the natural history of HPV infection and CIN2+/CIN3+ in a prospective cohort of WLHIV followed over a median 16 months in Burkina Faso (BF) and South Africa (SA) [16,17]. The primary objective of the current study was to evaluate the diagnostic accuracy of 3 screening approaches (index tests): HR-HPV DNA tests (Hybrid Capture 2 [HC2] and careHPV), visual inspection (standard of care in BF), and cervical cytology (standard of care in SA) for the detection of prevalent histology-confirmed CIN2+/CIN3+ (reference method) in screening and in triage (Analysis 1). Secondary objectives were to evaluate the diagnostic accuracy of those test strategies by ART status and age (Analysis 2). To inform on frequency of screening, we evaluated CIN2+ incidence over a median 16 months among baseline screen-negative WLHIV (Analysis 3).

## Methods

### Study design and participants

We enrolled WLHIV recruited from the Hôpital de Jour (the HIV outpatient clinic of the Internal Medicine Department at Centre Hospitalier Universitaire Yalgado [CHU-Yalgado]), Ouagadougou, BF, and the Esselen Street Clinic (a primary health clinic) and Ward 21 of Hillbrow Community Health Centre (an ART initiation site) in Johannesburg, SA, from 5

December 2011 to 30 October 2012 in a prospective evaluation study of cervical cancer screening strategies, as previously described [17]. In brief, women were enrolled consecutively in the HARP (HPV in Africa Research Partnership) study if they were HIV-1 seropositive, aged 25–50 years, and resident in the study city. Women were excluded if they had a history of prior treatment for cervical cancer, had previous hysterectomy, or were pregnant or less than 8 weeks postpartum. Enrolment was stratified in a 2:1 ratio of ART users:ART-naïve WLHIV. Participants were followed up every 6 months for CD4+ T lymphocyte cell count monitoring and up to month 18, when procedures similar to baseline were repeated (median 16 months after baseline). All women provided written informed consent. Ethical approval was granted by the Ministry of Health in BF (no. 2012-12-089), the University of the Witwatersrand in SA (no. 110707), and the London School of Hygiene & Tropical Medicine (no. 7400).

## Procedures

At baseline and endline (median 16 months later) visits, cervical samples were collected from all women using a Digene cervical sampler (Qiagen, Courtaboeuf, France) for HPV DNA testing and genotyping, a cytobrush for Papanicolaou smear cytology, and swabs from the ecto/endocervix and vagina to detect sexually transmitted infections (STIs). All participants had a visual inspection using acetic acid (VIA) and a visual inspection using Lugol's iodine (VILI) performed by trained nurses following the International Agency for Research on Cancer (IARC) guidelines [18]. All participants were referred for colposcopy at a median of 12 weeks (interquartile range [IQR] 8–15) following the baseline visit, performed by trained colposcopists applying the Swede score for clinical severity grading [19]. Colposcopists were aware of VIA/VILI, cytology, and HPV DNA test results. Systematic 4-quadrant cervical biopsy, including directed biopsy of any suspicious lesions, was performed for participants who had abnormalities detected by cytology (atypical squamous cells of undetermined significance, or greater [ASCUS+]) or VIA/VILI or during colposcopy, or who were HR-HPV DNA positive. A venous blood sample was collected to confirm HIV-1 serostatus if needed, and to obtain HIV-1 RNA plasma viral load and CD4+ T cell count.

HR-HPV testing using the qualitative Digene HC2, which detects 13 HR-HPV types (HPV16, -18, -31, -33, -35, -39, -45, -51, -52, -56, -58, -59, and -68), at baseline was performed centrally at the University of Montpellier (UM) virology laboratory by trained laboratory technicians in France as previously described [20]. The qualitative careHPV (Qiagen, Gaithersburg, MD), which detects 14 HR-HPV types (HPV16, -18, -31, -33, -35, -39, -45, -51, -52, -56, -58, -59, -66, and -68), was performed at endline by trained laboratory technicians at the Centre de Recherche Biomoléculaire Pietro Annigoni (CERBA), Ouagadougou, BF, and the National Health Laboratory Service (NHLS), Johannesburg, SA. A high level of agreement between HC2 and careHPV was reported in a nested study [21]. Quality assurance was performed by the UM virology laboratory. Results were displayed by the careHPV test controller without additional specification of the luminescent signal intensity. Genotyping with the INNO-LiPA HPV Genotyping Extra assay (Innogenetics, Courtaboeuf, France) was conducted at UM as previously described [20]. Conventional cytological reading was based on the Papanicolaou method and performed at the pathology department at CHU-Yalgado in Ouagadougou and the NHLS in Johannesburg according to the Bethesda classification system [22], with a quality assurance scheme organised by the UM virology laboratory for both countries. The NHLS lab was also subscribed to the Cytopathology Quality Assurance Program of the Royal College of Pathologists of Australasia Quality Assurance Program.

Cervical biopsies were processed at the local pathology laboratories and read using the 3-tier CIN classification system [23]. The reference standard of histology was classified as

'negative' (≤CIN1) or 'positive' (CIN2+) based on the highest reading across all findings from the 4-quadrant biopsies and endocervical curettage if collected. The histopathologist was blind to VIA/VILI, cytology, and HPV DNA test results but was aware of colposcopy diagnosis. All histological slides from women with a local diagnosis of CIN2+ and approximately 10% of slides from women with ≤CIN1 histological findings were reviewed by the HARP Endpoint Committee of 5 pathologists, for consensus classification, which showed high agreement [24].

Participants were recalled for CIN2+ management according to local guidelines at each site, if found to have CIN2+ lesions by histology at the baseline and/or endline visit. The management visit was scheduled at the earliest convenient date once the result was known. Due to demands on local health services in SA, this often meant that CIN2+ management was scheduled up to 14 months after diagnosis. CIN2+ status was therefore defined according to whether the participant had received CIN2+ management between enrolment and follow-up. CIN2 + prevalence at baseline was defined as the number of women with CIN2+ detected at baseline among all women enrolled in the HARP study. Cumulative CIN2+ prevalence at endline was defined as the number of women with CIN2+ detected at endline among all women attending the endline visit, irrespective of whether women were treated for prevalent CIN2+ between baseline and endline. CIN2+ incidence was defined as newly detected CIN2+ at the endline visit among women without CIN2+ at baseline.

## Statistical analysis

In the analysis of diagnostic accuracy (Analysis 1), the index tests evaluated included VIA alone, VIA/VILI (co-testing when either test is positive among all women screened), cytology (using thresholds of low-grade squamous intraepithelial lesion or greater [LSIL+] and high-grade squamous intraepithelial lesion or greater [HSIL+]), and HR-HPV DNA (Digene HC2) for the detection of histology-confirmed CIN2+ and CIN3+ (reference method) at baseline. Sensitivity, specificity, positive predictive value (PPV) complement of NPV (1 − NPV), the number of referrals to colposcopy that would be generated for each CIN2+ or CIN3+ case identified (number needed to refer [NNR] = 1/PPV) [25], and the number of referrals per 1,000 women screened were reported for each of the index tests. For HC2, we considered test positivity at varying thresholds of the relative light unit (RLU) between ≥1 and ≥20, corresponding with increasing HPV viral load [26], to evaluate the threshold effect on test specificity to distinguish CIN2+/CIN3+. Among HR-HPV (HC2) positive women, we evaluated the diagnostic accuracy of triage approaches, including VIA, VIA/VILI, cytology (ASCUS+ and HSIL+), a combination of HPV16/18 genotyping and cytology (test positive if HPV16 or HPV18 positive, or cytology [ASCUS+ or HSIL+] when negative for both HPV16 and HPV18), and combination of HPV16/18 genotyping and VIA (test positive if HPV16 or HPV18 positive, or VIA abnormal when negative for both HPV16 and HPV18).

We evaluated the diagnostic accuracy of a restricted genotyping approach using results of the INNO-LiPA genotyping assay in the following combinations (positive for any genotype): HPV16; HPV16/18/45 (3 high-risk [HR] types), and HPV16/18/45/31/33/35/52/58 (8 HR types). Because of the low limit of detection of INNO-LiPA and to improve clinical relevancy, test positivity was defined as positivity for any of those genotypes among women who were also HC2 positive. We also evaluated the diagnostic accuracy of an HPV-based test targeting HR types previously reported to be most significantly associated with CIN3+ in the HARP cohort [27]. Relative sensitivity (RSen) and relative specificity (RSpec) and 95% confidence intervals (CIs) of screening tests compared to the standard of care in each country (VIA/VILI in BF and HSIL+ cytology in SA) were calculated [28].

In order to observe the performance of screening strategies in an already screened population, we evaluated the diagnostic accuracy of endline VIA/VILI, HPV DNA (careHPV), and

cytology for cumulative CIN2+ detection at endline, excluding women who were treated for prevalent CIN2+ at baseline.

To evaluate the association of HIV-related factors with diagnostic accuracy of screening strategies, diagnostic accuracy for CIN2+/CIN3+ was evaluated separately among women on prolonged ART (>2 years), women on short-duration ART (≤2 years), and ART-naïve women at baseline (Analysis 2). Diagnostic accuracy was also evaluated according to age at screening (Analysis 2). The cumulative incidence of CIN2+ at endline was calculated among women who screened negative for each of the screening strategies at baseline (Analysis 3). Analyses for diagnostic accuracy were conducted for discrete outcomes of CIN2+ and CIN3+. Data are presented separately for each country. Data were analysed using Stata (version 16) and according to the study statistical analysis plan (S1 Text). This article was reported according to the Standards for Reporting of Diagnostic Accuracy (STARD) statement (S1 Checklist) [29]. The dataset is available in the Mendeley Data online repository at doi: 10.17632/yd5ygw38vj.1.

## Results

### Study population

Of 1,395 women screened, 1,238 (89%) were enrolled in the HARP study (BF: 615; SA: 623; Fig 1). Overall, 1,130 (91.3%) participants (BF: 90.1%; SA: 92.5%) had valid histology and were included in the final analysis. The median time from enrolment, when index tests were conducted, to the colposcopy visit, when biopsy was taken, if indicated, for histology verification, was 2.9 months (IQR 2.1–3.8). CIN2+ prevalence was 5.8% (32/554) in BF and 22.4% (129/576) in SA ($p < 0.001$; Table 1). CIN3+ prevalence was 2.3% (13/554) in BF and 9.2% (53/576) in SA ($p < 0.001$).

The median age of participants was 36 (IQR 32–42) years in BF and 34 (IQR 30–39) years in SA (Table 1). About half (49.0%) of SA participants had ever had a Pap smear, and a fifth (21.5%) of BF participants had ever had a VIA/VILI examination, the respective primary cervical cancer screening modality in each country. At enrolment, 387 (69.9%) participants were on ART in BF and 370 (64.2%) in SA ($p = 0.045$), reflecting the 2:1 stratification ratio. Just over a third of women were taking ART for >2 years. The median CD4+ T cell count among women on ART for a prolonged duration (>2 years), women on ART for a short duration ART (≤2 years), and ART-naïve women was 478 (IQR 366–478), 394 (IQR 276–573), and 392 (IQR 310–591) cells/μl, respectively, in BF and 476 (IQR 372–626), 326 (IQR 207–453), and 440 (IQR 347–595) cells/μl, respectively, in SA. The corresponding values for HIV-1 plasma viral load were 40 (IQR 40–40), 40 (IQR 40–101), and 23,171 (IQR 1,943–166,067) copies/ml, respectively, in BF and 109 (IQR 40–560), 141 (IQR 40–743), and 19,650 (IQR 4,800–57,500) copies/ml, respectively, in SA.

The prevalence of HR-HPV by INNO-LiPA genotyping was 57.7% (315/556) in BF and 79.3% (456/575) in SA ($p < 0.001$). The prevalence of HR-HPV by HC2 was 41.8% (229/554) in BF and 59.7% (342/576) in SA (Table 1).

### Diagnostic accuracy of screening strategies for CIN2+ and CIN3+ detection at baseline

At baseline, positivity for each of the screening tests—VIA, VIA/VILI, ASCUS+, HSIL+, and HR-HPV DNA (HC2)—was 21.0%, 23.9%, 25.8%, 4.5%, and 41.8%, respectively, in BF, and 28.1%, 41.5%, 93.4%, 30.1%, and 59.7%, respectively, in SA (Table 1).

VIA had low sensitivity for CIN2+/CIN3+ (countries combined—CIN2+: 44.7%, 95% CI 36.9%–52.7%; CIN3+: 56.1%, 95% CI 43.3%–68.3%), with specificity for ≤CIN1 of 78.7%

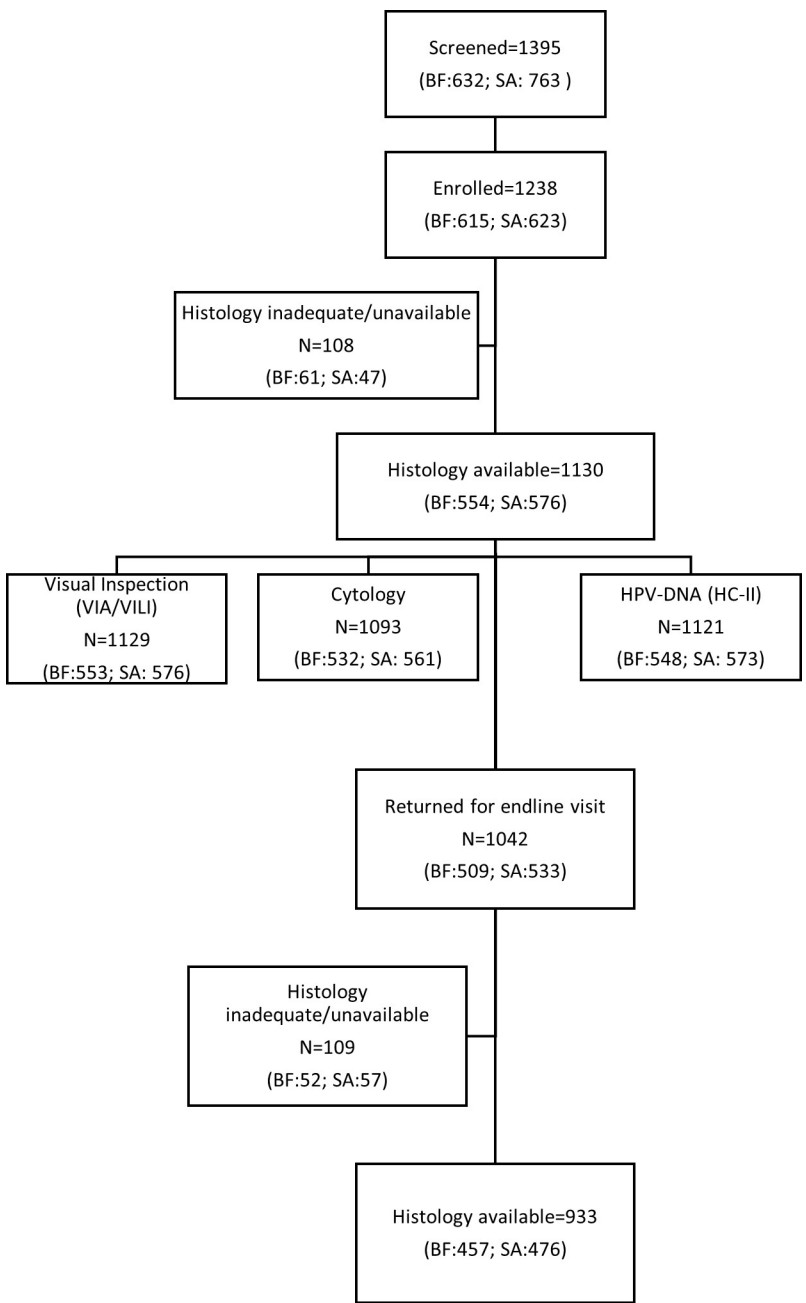

**Fig 1. Study flowchart.** BF, Burkina Faso; HC-II, Hybrid Capture 2; HPV, human papillomavirus; SA, South Africa; VIA, visual inspection using acetic acid; VILI, visual inspection using Lugol's iodine.

(95% CI 76.0%–81.3%; Tables 2 and 3). The addition of VILI to VIA (either positive) resulted in an increase in sensitivity (CIN2+: 61.5%, 95% CI 53.5%–69.0%; CIN3+: 69.7%, 57.1%–80.4%), with highest sensitivity observed for CIN3+ in BF only (84.6%, 95% CI 54.6%–98.1%; S1 Table; Fig 2). The number of referrals to colposcopy varied by country for VIA/VILI: 239 women per 1,000 women screened in BF and 415 in SA (S2 Table). The PPV varied by country, reflecting the difference in CIN3+ prevalence: 8.3% (95% CI 4.2%–14.4%) and 14.6% (95% CI 10.4%–19.8%) in BF and SA, respectively.

**Table 1. Patient characteristics and screening test positivity at baseline and endline in Burkina Faso and South Africa.**

| Characteristic or screening test positivity | Burkina Faso | | South Africa | | p-Value |
|---|---|---|---|---|---|
| | Baseline N = 554 | Endline N = 457 | Baseline N = 576 | Endline N = 476 | |
| **Baseline characteristics** | | | | | |
| Age, median (IQR) | 36 (32, 42) | — | 34 (30, 39) | — | <0.001 |
| Ever had prior cervical cancer screening | 158 (28.6) | — | 285 (49.6) | — | <0.001 |
| Ever had Pap smear | 69 (12.5) | — | 282 (49.0) | — | <0.001 |
| Ever had visual inspection exam | 119 (21.5) | — | 15 (2.6) | — | <0.001 |
| Taking hormonal contraception | 290 (52.4) | — | 492 (85.4) | — | <0.001 |
| Years since HIV diagnosis | | — | | — | |
| <5 years | 258 (46.6) | — | 325 (56.4) | — | 0.003 |
| 5–9 years | 226 (40.8) | — | 186 (32.3) | — | |
| ≥10 years | 70 (12.6) | — | 65 (11.3) | — | |
| ART status | | — | | — | |
| ART >2 years | 220 (39.7) | — | 207 (35.9) | — | 0.129 |
| ART ≤2 years | 167 (30.1) | — | 163 (28.3) | — | |
| ART naïve | 167 (30.1) | — | 206 (35.8) | — | |
| HIV-1 PVL[1] among ART-naïve women, median (IQR) | 14,944 (40, 128,897) | — | 18,150 (3,800, 55,400) | — | <0.001 |
| HIV-1 PVL suppression[2] among ART users | 336 (86.8) | — | 299 (80.8) | — | |
| **Screen test positive** | | | | | |
| VIA only positive | 116 (21.0) | 62 (13.6) | 162 (28.1) | 95 (20.0) | |
| VILI only positive | 130 (23.5) | 68 (15.1) | 219 (38.0) | 208 (43.7) | |
| VIA or VILI positive (VIA/VILI) | 132 (23.9) | 76 (16.6) | 239 (41.5) | 220 (46.2) | |
| Cytology ASCUS+ | 137 (25.8) | 45 (10.9) | 524 (93.4) | 459 (97.3) | |
| Cytology LSIL+ | 120 (22.6) | 34 (8.2) | 504 (89.8) | 378 (80.1) | |
| Cytology HSIL+ | 24 (4.5) | 7 (1.7) | 169 (30.1) | 85 (18.0) | |
| HR-HPV DNA positive[3] | 229 (41.8) | 190 (41.9) | 342 (59.7) | 281 (59.3) | |
| Histology-confirmed grade | | | | | |
| <CIN1 | 373 (67.3) | 383 (83.8) | 262 (45.5) | 397 (83.4) | |
| CIN1 | 149 (26.9) | 68 (14.9) | 185 (32.1) | 32 (6.7) | |
| CIN2 | 19 (3.4) | 5 (1.1) | 76 (13.2) | 34 (7.1) | |
| CIN3 | 11 (2.0) | 1 (0.2) | 53 (9.2) | 13 (2.7) | |
| Invasive cervical cancer | 2 (0.4) | 0 (0.0) | 0 (0.0) | 0 (0.0) | |
| ≤CIN1 | 522 (94.2) | 451 (98.7) | 447 (77.6) | 429 (90.1) | |
| CIN2+ | 32 (5.8) | 6 (1.3) | 129 (22.4) | 47 (9.9) | |
| CIN3+ | 13 (2.3) | 1 (0.2) | 53 (9.2) | 13 (2.7) | |

Data are n (%) unless otherwise indicated.

[1] RNA, copies/ml.

[2] PVL < 1,000 copies/ml.

[3] Using Hybrid Capture 2 at baseline (cutoff of 1 relative light unit) and careHPV at endline.

ASCUS+, atypical squamous cells of undetermined significance, or greater; CIN, cervical intraepithelial neoplasia; HR-HPV, high-risk human papillomavirus; HSIL, high-grade squamous intraepithelial lesion; LSIL, low-grade squamous intraepithelial lesion; PVL, plasma viral load; VIA, visual inspection using acetic acid; VILI, visual inspection using Lugol's iodine.

Diagnostic accuracy of cervical cytology varied by country. In SA, cytology using a cutoff of HSIL+ had the best combination of sensitivity (CIN2+: 70.1%, 95% CI 61.3%–77.9%; CIN3+: 80.8%, 95% CI 67.5%–90.4%) and specificity for ≤CIN1 (81.6%, 95% CI 77.6%–85.1%; Fig 3) and would result in 301 referrals per 1,000 women, with a PPV for CIN3+ of 24.9% (95% CI

**Table 2. Performance of screening strategies for detection of prevalent CIN2+ among 1,130 women living with HIV (554 in BF; 576 in SA).**

| Strategy | Tests performed, n | Test positive (colposcopy indicated), n (%) | Number of colposcopies per 1,000 women screened | Number of colposcopies needed to detect 1 case of CIN2+, n | Sensitivity percent (95% CI) | Specificity (95% CI) | PPV (95% CI) | 1 − NPV (95% CI) | AUC (95% CI) |
|---|---|---|---|---|---|---|---|---|---|
| **Stand-alone tests** | | | | | | | | | |
| VIA positive | 1,129 | 278 (24.6) | 246 | 3.9 | 44.7 (36.9–52.7) | 78.7 (76.0–81.3) | 25.9 (20.9–31.5) | 10.5 (8.5–12.7) | 0.62 (0.58–0.66) |
| VIA or VILI positive (VIA/VILI) | 1,129 | 371 (32.9) | 329 | 3.7 | 61.5 (53.5–69.0) | 71.9 (69.0–74.7) | 26.7 (22.3–31.5) | 8.2 (6.3–10.4) | 0.67 (0.63–0.71) |
| Cytology ASCUS+ (BF) | 532 | 137 (25.8) | 258 | 6.0 | 76.7 (57.7–90.1) | 77.3 (73.4–80.9) | 16.8 (11.0–24.1) | 1.8 (0.7–3.6) | 0.77 (0.69–0.85) |
| Cytology HSIL+ (SA) | 561 | 169 (30.1) | 301 | 1.9 | 70.1 (61.3–77.9) | 81.6 (77.6–85.1) | 52.7 (44.9–60.4) | 9.7 (7.0–13.1) | 0.76 (0.71–0.80) |
| HC2 (≥1 RLU) | 1,121 | 571 (51.3) | 513 | 4.0 | 88.8 (82.9–93.2) | 55.4 (52.2–58.6) | 25.0 (21.5–28.8) | 3.3 (2.0–5.1) | 0.72 (0.69–0.75) |
| HC2 (≥10 RLU) | 1,121 | 445 (39.7) | 397 | 3.5 | 80.1 (73.1–86.0) | 67.1 (64.0–70.1) | 29.0 (24.8–33.4) | 4.7 (3.3–6.6) | 0.74 (0.70–0.77) |
| HC2 (≥20 RLU) | 1,121 | 398 (35.5) | 355 | 3.2 | 76.4 (69.1–82.7) | 71.4 (68.4–74.2) | 30.9 (26.4–35.7) | 5.3 (3.7–7.1) | 0.74 (0.70–0.78) |
| 8 HR types[1] | 1,121 | 378 (33.7) | 337 | 3.1 | 76.9 (69.6–83.2) | 73.5 (70.6–76.2) | 32.5 (27.8–37.5) | 5.0 (3.5–6.8) | 0.75 (0.72–0.79) |
| **Triage of HPV-positive women[2]** | | | | | | | | | |
| VIA positive | 571 | 175 (30.6) | 156 | 2.7 | 45.5 (37.1–54.0) | 74.3 (69.9–78.4) | 37.1 (30.0–44.8) | 19.7 (15.9–24.0) | 0.60 (0.55–0.65) |
| VIA or VILI positive (VIA/VILI) | 571 | 237 (41.5) | 211 | 2.6 | 62.9 (54.5–70.9) | 65.7 (60.9–70.1) | 38.0 (31.8–44.5) | 15.9 (12.1–20.8) | 0.64 (0.60–.69) |
| Cytology ASCUS+ (BF) | 217 | 90 (41.5) | 173 | 4.1 | 75.9 (56.5–89.7) | 63.8 (56.5–70.7) | 24.4 (16.0–34.6) | 5.5 (92.2–11.0) | 0.70 (0.61–0.79) |
| Cytology HSIL+ (SA) | 333 | 150 (45.0) | 269 | 1.8 | 74.8 (65.6–82.5) | 69.8 (63.3–75.8) | 55.3 (47.0–63.4) | 15.3 (10.4–21.3) | 0.72 (0.67–0.77) |
| HPV16/18+ or other HR-HPV + and reflex HSIL+[3] | 555 | 277 (49.9) | 254 | 2.5 | 80.7 (73.2–86.9) | 60.5 (55.6–65.2) | 40.8 (35.0–46.8) | 9.7 (6.5–13.8) | 0.71 (0.67–0.75) |
| HPV16/18+ or other HR-HPV + and reflex VIA[4] | 570 | 285 (50.0) | 255 | 2.9 | 69.0 (60.7–76.5) | 56.3 (51.5–61.1) | 34.4 (28.9–40.2) | 5.4 (11.4–20.2) | 0.63 (0.58–0.67) |

[1]Positive for HC2 (using a threshold of ≥10 RLU) and any of HPV16/18/31/33/35/45/52/58.

[2]Calculated among women testing positive for HPV DNA, using HC2 ≥ 1 RLU to define test positive.

[3]Test positive if HPV16 or HPV18 positive, or cytology (HSIL+) when negative for both HPV16 and HPV18.

[4]Test positive if HPV16 or HPV18 positive, or VIA abnormal when negative for both HPV16 and HPV18.

ASCUS+, atypical squamous cells of undetermined significance, or greater; AUC, area under the curve; BF, Burkina Faso; CIN, cervical intraepithelial neoplasia; HC2, Hybrid Capture 2; HPV, human papillomavirus; HR, high risk; HR-HPV, high-risk human papillomavirus; HSIL+, high-grade squamous intraepithelial lesion or greater; NPV, negative predictive value; PPV, positive predictive value; RLU, relative light unit; SA, South Africa; VIA, visual inspection using acetic acid; VILI, visual inspection using Lugol's iodine.

**Table 3. Performance of screening strategies for detection of prevalent CIN3+ among 1,130 women living with HIV (554 in BF; 576 in SA).**

| Strategy | Tests performed, n | Number of colposcopies needed to detect 1 case of CIN3+, n | Sensitivity percent (95% CI) | Specificity (95% CI) | PPV (95% CI) | 1 − NPV (95% CI) | AUC (95% CI) |
|---|---|---|---|---|---|---|---|
| **Stand-alone tests** | | | | | | | |
| VIA positive | 1,129 | 7.5 | 56.1 (43.3–68.3) | 77.3 (74.7–79.8) | 13.3 (9.6–17.9) | 3.4 (2.3–4.9) | 0.67 (0.61–0.73) |
| VIA or VILI positive (VIA/VILI) | 1,129 | 8.1 | 69.7 (57.1–80.4) | 69.4 (66.6–72.2) | 12.4 (9.2–16.2) | 2.6 (1.6–4.0) | 0.70 (0.64–0.75) |
| Cytology ASCUS+ (BF) | 532 | 17.1 | 72.7 (39.0–94.0) | 75.2 (71.3–78.9) | 5.8 (2.6–11.2) | 0.8 (0.2–2.2) | 0.74 (0.60–0.88) |
| Cytology HSIL+ (SA) | 561 | 4.0 | 80.8 (67.5–90.4) | 75.0 (71.1–78.8) | 24.9 (18.5–32.1) | 2.6 (1.2–4.6) | 0.78 (0.72–0.84) |
| HC2 ($\geq$1 RLU) | 1,121 | 10.0 | 86.4 (75.7–93.6) | 51.3 (48.2–54.3) | 10.0 (7.7–12.7) | 1.6 (0.8–3.1) | 0.69 (0.64–0.73) |
| HC2 ($\geq$10 RLU) | 1,121 | 8.6 | 78.8 (67.0–87.9) | 62.7 (59.8–65.7) | 11.7 (8.9–15.0) | 2.1 (1.1–3.5) | 0.71 (0.66–0.76) |
| HC2 ($\geq$20 RLU) | 1,121 | 8.0 | 75.8 (63.6–85.5) | 67.0 (64.1–69.8) | 12.6 (9.5–16.2) | 2.2 (1.3–3.6) | 0.71 (0.66–0.77) |
| 8 HR types[1] | 1,121 | 7.4 | 77.3 (65.3–86.7) | 69.0 (66.1–71.8) | 13.5 (10.2–17.4) | 2.0 (1.1–3.3) | 0.73 (0.68–0.78) |
| **Triage of HPV-positive women[2]** | | | | | | | |
| VIA positive | 571 | 5.5 | 56.1 (42.4–69.3) | 72.2 (68.1–76.0) | 18.3 (12.9–24.8) | 6.3 (4.1–9.2) | 0.64 (0.57–0.71) |
| VIA or VILI positive (VIA/VILI) | 571 | 5.9 | 70.2 (56.6–81.6) | 61.7 (57.3–65.9) | 16.9 (12.3–22.3) | 5.1 (3.0–8.0) | 0.66 (0.60–0.72) |
| Cytology ASCUS+ (BF) | 217 | 11.3 | 72.7 (39.0–94.0) | 60.2 (53.2–66.9) | 8.9 (3.9–16.8) | 2.4 (0.5–6.7) | 0.67 (0.52–0.81) |
| Cytology HSIL+ (SA) | 333 | 3.9 | 88.4 (74.9–96.1) | 61.4 (55.5–67.0) | 25.3 (18.6–33.1) | 2.7 (0.9–6.3) | 0.75 (0.69–0.81) |
| HPV16/18+ or other HR-HPV+ and reflex HSIL+[3] | 555 | 5.5 | 90.9 (80.0–97.0) | 54.6 (50.1–59.0) | 18.1 (13.7–23.1) | 1.8 (0.6–4.1) | 0.73 (0.68–0.77) |
| HPV16/18+ or other HR-HPV+ and reflex VIA[4] | 570 | 6.5 | 77.2 (64.2–87.3) | 53.0 (48.6–57.4) | 15.4 (11.4–20.2) | 4.6 (2.5–7.7) | 0.65 (0.59–0.71) |

[1]Positive for HC2 (using a threshold of $\geq$10 RLU) and any of HPV16/18/31/33/35/45/52/58.

[2]Calculated among women testing positive for HPV DNA, using HC2 $\geq$ 1 RLU to define test positive.

[3]Test positive if HPV16 or HPV18 positive, or cytology (HSIL+) when negative for both HPV16 and HPV18.

[4]Test positive if HPV16 or HPV18 positive, or VIA abnormal when negative for both HPV16 and HPV18.

ASCUS+, atypical squamous cells of undetermined significance, or greater; AUC, area under the curve; BF, Burkina Faso; CIN, cervical intraepithelial neoplasia; HC2, Hybrid Capture 2; HPV, human papillomavirus; HR, high risk; HR-HPV, high-risk human papillomavirus; HSIL+, high-grade squamous intraepithelial lesion or greater; NPV, negative predictive value; PPV, positive predictive value; RLU, relative light unit; SA, South Africa; VIA, visual inspection using acetic acid; VILI, visual inspection using Lugol's iodine.

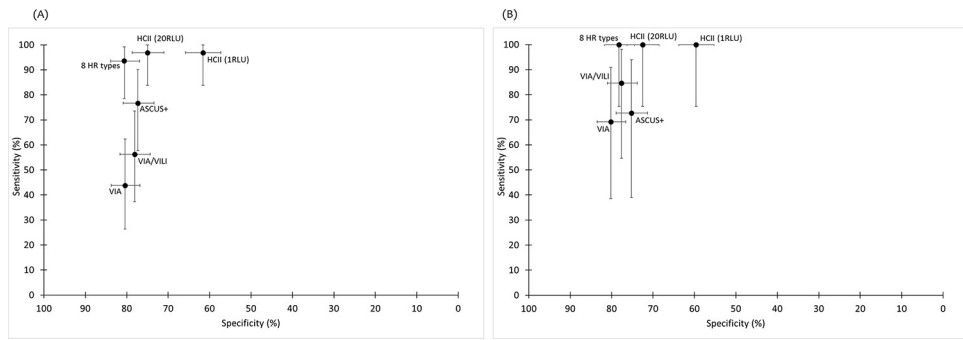

**Fig 2. Sensitivity and specificity of screening strategies for prevalent CIN2+ and CIN3+ in Burkina Faso.** (A) CIN2 +; (B) CIN3+. ASCUS+, atypical squamous cells of undetermined significance, or greater; CIN, cervical intraepithelial neoplasia; HCII, Hybrid Capture 2; HR, high risk; RLU, relative light unit; VIA, visual inspection using acetic acid; VILI, visual inspection using Lugol's iodine.

18.5%–32.1%; Table 3). In BF, a cutoff of ASCUS+ had an optimal combination of sensitivity for CIN3+ of 72.7% (95% CI 39.0%–94.0%) and specificity for ≤CIN1 of 77.3% (95% CI 73.4%–80.9%).

HC2 using a threshold of ≥1 RLU had the highest sensitivity of all screening strategies for CIN2+ (88.8%, 95% CI 82.9%–93.2%) and CIN3+ (86.4%, 95% CI 75.7%–93.6%) but the lowest specificity (55.4%, 95% CI 52.2%–58.6%; Tables 2 and 3), and the proportion of women testing positive was 51.3% (41.8% in BF and 59.7% in SA, $p < 0.001$). Increasing the threshold to ≥20 RLU resulted in fewer women screening positive (35.5% overall, 29.2% in BF and 41.5% in SA) and increased the specificity to 71.4% (95% CI 68.4%–74.2%) but with loss in sensitivity (CIN2+: 76.4%, 95% CI 69.1%–82.7%; CIN3+: 75.8%, 95% CI 63.6%–85.5%). In BF, the number of colposcopy referrals was similar to current standard-of-care VIA/VILI (292 versus 239 per 1,000 women screened for HC2 [≥20 RLU] and VIA/VILI, respectively), with a 2-fold increase in sensitivity for CIN2+ (96.9% versus 56.3%; RSen = 1.72, 95% CI 1.28–2.32; S3 Table) but only marginally greater sensitivity for CIN3+ (RSen = 1.18, 95% CI 0.94–1.49). In SA, HC2 (≥1 RLU) had similar sensitivity for CIN3+ as HSIL+ (83.0%, 95% CI 70.2%– 91.9%; RSen = 1.02, 95% CI 0.89–1.18) but lower specificity (42.7%, 95% CI 38.4%–47.1%; RSpec = 0.57, 95% CI 0.52–0.63), resulting in almost twice as many referrals (597 per 1,000 women) and lower PPV compared to HSIL+ (12.9%, 95% CI 9.5%–16.9%). Increasing the

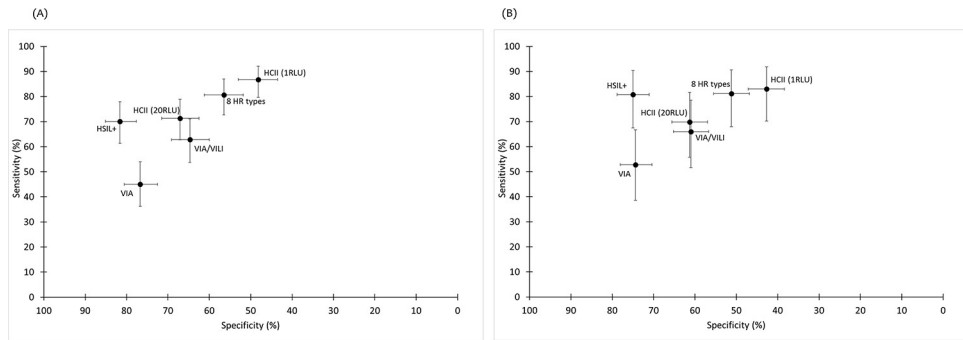

**Fig 3. Sensitivity and specificity of screening strategies for prevalent CIN2+ and CIN3+ in South Africa.** (A) CIN2 +; (B) CIN3+. CIN, cervical intraepithelial neoplasia; HCII, Hybrid Capture 2; HR, high risk; HSIL+, high-grade squamous intraepithelial lesion or greater; RLU, relative light unit; VIA, visual inspection using acetic acid; VILI, visual inspection using Lugol's iodine.

threshold to define test positivity decreased the number of referrals in SA but was associated with a loss in sensitivity (S2 Table).

Using a combination of increased threshold and a restricted genotype approach targeting 8 HR types resulted in the best combination of sensitivity for CIN3+ (countries combined: 77.3%, 95% CI 65.3%–86.7%; Table 3) and specificity for ≤CIN1 (73.5%, 95% CI 70.6%–76.2%; Table 2) of any of the HPV-based strategies. Higher sensitivity and specificity for CIN3+ were observed in BF using this approach (100.0% and 78.2%, respectively; S1 Table).

In both countries, triage of HC2-positive (≥1 RLU) women using VIA or VIA/VILI had similarly low sensitivity for CIN2+ as using VIA or VIA/VILI as a screen test. In BF, although triage of HC2-positive (≥20 RLU) women with VIA/VILI had low sensitivity for CIN2+ (58.1%, 95% CI 39.1%–75.5%; S3 Table), this approach had high sensitivity for CIN3+ (84.6%, 95% CI 54.6%–98.1%; S1 Table) and decreased the number of colposcopy referrals to 100 per 1,000 women. In SA, triage of HC2-positive (≥1 RLU) women with cytology HSIL+ increased the number of referrals compared to HSIL+ alone, from 301 to 363 per 1,000 women, but referral rate was 40% lower than using HC2 alone (597 per 1,000 women). Although sensitivity for CIN3+ was high (88.4%, 95% CI 74.9%–96.1%; S2 Table; S4 Table), this approach would miss 26.9% (14/52) of all women with CIN3+ in SA due to the lower sensitivity of HC2 as a screen test in SA compared to BF. Triage of HC2-positive women with a combination of HPV16/18 and HSIL+ had higher sensitivity (95.3%, 95% CI 84.2%–99.4%), with a marginally higher number of colposcopy referrals (370 per 1,000 women; S2 Table).

## Diagnostic accuracy of screening strategies by age at screening

The specificity of HPV tests for ≤CIN1 increased with increasing age (43.5%, 51.6%, 62.0%, 60.2%, and 63.0% in women aged 25–29, 30–34, 35–39, 40–44, and 45–50 years, respectively; S5 Table). The PPV increased and test positivity decreased with increasing age, corresponding with lower HR-HPV prevalence in older age groups. HC2 test positivity was highest and specificity was lowest in women aged 25–29 years. Triage of women aged 25–29 years using HPV16/18 with reflex cytology HSIL+ of non-HPV16/18 types generated a sensitivity in triage for CIN3+ of 100.0% (95% CI 73.5%–100.0%) and specificity of 60.5% (95% CI 51.1%–69.3%), and triage HPV16/18 with reflex VIA generated a sensitivity and specificity of 91.7% (95% CI 61.5%–99.8%) and 50.0% (95% CI 40.7%–59.3%), respectively (sites combined).

## The role of HIV-related factors in diagnostic accuracy of screening strategies

In both countries, HR-HPV prevalence was higher among ART-naïve women (58.8%) and recent ART users (≤2 years' duration: 60.6%) compared to prolonged ART users (>2 years: 40.0%; $p < 0.001$), as was CIN3+ prevalence (5.6%, 8.5%, and 4.5%, respectively, $p = 0.064$).

The sensitivity of VIA for CIN3+ was lower in women on ART >2 years (42.1%, 95% CI 20.3%–66.5%) compared to women on ART ≤2 years (64.3%, 95% CI 44.1%–81.4%) or ART-naïve (55.6%, 95% CI 30.8%–78.5%; S6 Table).

Specificity of HC2 (≥1 RLU) for ≤CIN1 was higher in women on ART >2 years (65.4%, 95% CI 60.3%–70.1%) compared to women on ART ≤2 years (47.6%, 95% CI 41.4%–53.7%) or ART-naïve women (46.2%, 95% CI 40.1%–52.4%), corresponding with lower HR-HPV prevalence in prolonged ART users (S7 Table). Consequently, a higher number of colposcopy referrals was observed among women on ART ≤2 years and ART-naïve women. In BF, VIA/VILI triage of HC2-positive women who were ART naïve or recent ART users decreased the number of colposcopy referrals from 498 (HC2 alone) to 154 (HC2 followed by VIA/VILI) per 1,000 women, with a good combination of sensitivity and specificity for CIN3+ (85.7%, 95%

CI 42.1%–99.6%, and 71.9%, 95% CI 63.5%–79.2%). In SA among women taking ART, HSIL
+ was the best performing test for CIN2+ and CIN3+, irrespective of duration of use. Among
ART-naïve women, however, the sensitivity of HSIL+ was low (CIN2+: 53.1%, 95% CI 38.3%–
67.5%; CIN3+: 66.7%, 95% CI 41.0%–86.7%).

### HR-HPV type-specific persistence and CIN status over 16 months

Of the 1,130 women evaluated at baseline, 1,042 (92.4%) were seen at the endline visit, at a
median follow-up of 16.2 months (IQR 15.6–16.8), of whom 933 (89.5%) had histology data
available at both time points (BF: 457; SA: 476; Fig 1). The cumulative prevalence of CIN2
+ was 1.3% (6/457) in BF and 9.9% (47/476) in SA ($p < 0.001$; Table 1), and of CIN3+ was
0.2% (1/457) in BF and 2.7% (13/476) in SA ($p < 0.001$). There were no invasive cancer cases
detected in either country at endline. Among 809 participants without CIN2+ at baseline, the
incidence of CIN2+ over 16 months was 3.3% (95% CI 2.3%–4.8%) overall and was higher in
SA (BF: 1.2% [5/430]; SA: 5.8% [22/379]; $p < 0.001$).

At endline, 27 (84.4%) women in BF with CIN2+ detected at baseline who underwent man-
agement of their CIN2/3 lesions returned for the endline visit. In SA, 97 women with CIN2
+ detected at baseline returned for the endline visit, and of these, 61 (63%) underwent manage-
ment before the colposcopy/biopsy endline visit. Of the 36 participants who did not undergo
treatment, 20 (55.6%) had CIN2/3 detected again at endline, and 16 (44.4%) had lower grade
lesions (≤CIN1). Of the women who underwent management, the median time from colpos-
copy visit to management was similar in both countries (BF: 10.5 months, IQR 7.3–12.6; SA:
10.7 months, IQR 6.2–13.8).

There were 903 women with matched histology and genotyping at both time points. Type-
specific HR-HPV persistence was 20.7% (156/752) among women who were ≤CIN1 at both
time points and 77.8% (21/27) among women with incident CIN2+. Among 87 women who
received management for prevalent CIN2+, HR-HPV persistence was 37.0% (30/82) in those
who remained ≤CIN1 and 66.7% (4/6) in women with CIN2+ redetected at endline (S8
Table). Among 36 women who did not receive management for prevalent CIN2+, HR-HPV
persistence was 70.0% (14/20) in women who remained CIN2+ and 35.3% (6/16) in women
who were ≤CIN1 at endline ($p$-trend < 0.001). A test targeting HR-HPV type-specific persis-
tence could detect 73.6% (95% CI 59.7%–84.7%; 39/53) cumulative CIN2+ cases at endline
with a PPV of 16.9% (95% CI 12.3%–22.3%). The proportion of women without CIN2+ at end-
line and with HR-HPV type persistence was 22.6% (95% CI 19.8%–25.5%; 192/880).

### CIN2+ incidence at endline among screen-negative women at endline

CIN2+ incidence at endline was 0.5% (95% CI 0.1–1.8) among women with a baseline negative
HC2 (≥1 RLU) test or <LSIL on cytology, and 2.2% (95% CI 1.3%–3.7%) among women who
were baseline VIA negative (Table 4). Among HC2-positive women with a triage test (VIA
and HSIL+), incident CIN2+ was higher in the screen/triage-negative women (HC2 followed
by VIA: 2.1%; HC2 followed by HSIL+: 1.8%) compared to women who were negative using
HC2 alone, because of the lower sensitivity of VIA and HSIL+, compared to HC2, for CIN2+.

## Discussion

We evaluated the diagnostic accuracy of screen and screen–triage approaches for CIN2
+/CIN3+ in a large prospective cohort of WLHIV from 2 African countries with different HIV
epidemics, different burdens of HPV infection and cervical cancer, and differing approaches
to screening for cervical cancer. This allows the findings to be extended to a range of low- and
middle-income settings. We found that an HPV-DNA-based test had high sensitivity but low

**Table 4. CIN2+ incidence at endline among baseline screen-negative women (countries combined).**

| Strategy | N tested | Incident CIN2+ screen negative, n (%, 95% CI) | Incident CIN2+ screen positive, n (%, 95% CI) |
|---|---|---|---|
| VIA | 809 | 14 (2.2, 1.3–3.7) | 13 (7.6, 4.4–12.6) |
| VIA/VILI | 809 | 11 (1.9, 1.1–3.4) | 16 (7.0, 4.3–11.1) |
| LSIL+ | 779 | 2 (0.5, 0.1–2.1) | 25 (6.3, 4.3–9.2) |
| HSIL+ | 779 | 12 (1.7, 0.9–3.0) | 15 (19.0, 11.7–29.3) |
| HC2 (≥1 RLU) | 803 | 2 (0.5, 0.1–1.8) | 25 (6.9, 4.7–10.1) |
| HC2 (≥20 RLU) | 803 | 10 (1.7, 0.5–3.2) | 17 (7.5, 4.7–11.8) |
| 8 HR types[1] | 788 | 3 (0.6, 0.2–1.8) | 24 (8.5, 5.8–12.4) |
| HC2 (≥1 RLU) → VIA[2] | 807 | 15 (2.1, 1.3–3.5) | 12 (13.0, 7.5–21.7) |
| HC2 (≥20 RLU) → VIA[2] | 803 | 19 (2.6, 1.6–4.0) | 8 (12.3, 6.2–23.0) |
| HC2 (≥1 RLU) → HSIL+[2] | 787 | 13 (1.8, 1.0–3.1) | 14 (21.5, 13.1–33.4) |
| HC2 (≥20 RLU) → HSIL+[2] | 787 | 17 (2.3, 1.4–3.7) | 10 (18.2, 9.9–30.9) |

[1]Positive for HC2 (using a threshold of ≥1 RLU) and any HPV16/18/31/33/35/45/52/58.

[2]Incident CIN2+ in baseline screen-negative women calculated among women who were negative for either the screen or triage test at baseline (i.e., not restricted to screen-positive women to account for women with false-negative results in the initial screen test); incident CIN2+ in screen-positive women calculated among women who tested positive for both the screen and triage test.

CIN, cervical intraepithelial neoplasia; HC2, Hybrid Capture 2; HPV, human papillomavirus; HR, high risk; HSIL+, high-grade squamous intraepithelial lesion or greater; RLU, relative light unit; VIA, visual inspection using acetic acid; VILI, visual inspection using Lugol's iodine.

specificity for CIN2+/CIN3+, but with simple modifications to increase the threshold for test positivity and with a restricted genotype approach resulted in higher specificity and correspondingly fewer referrals to colposcopy. Triage of HPV-positive women with VIA/VILI in BF and cytology (HSIL+) in SA resulted in a further reduction in referrals, with minimal impact on sensitivity for CIN3+, but not for CIN2+.

HPV-based tests have high sensitivity for CIN2+/CIN3+ in both HIV-negative women and WLHIV, but specificity to distinguish CIN2+ is lower in WLHIV compared to HIV-negative women [30–34]. HPV-based tests targeting up to 14 HR types, including HC2, careHPV, and GeneXpert, have all shown high sensitivity but low specificity for CIN2+/CIN3+ in WLHIV [4–6,31,35,36], due to the high prevalence of HPV infection among these women. In a meta-analysis of 20 studies evaluating the association between HR-HPV prevalence and the specificity of HPV DNA testing (HC2) to distinguish CIN2+, HC2 specificity decreased by 8.4% (95% CI 8.02%–8.81%) for each 10% increase in HR-HPV prevalence [7]. In the HARP study, approximately half of the WLHIV with HR-HPV were infected with 2 or more HR types, and 19% were infected with 3 or more at baseline. Over 16 months, 35% of infections persisted, and 54% of women acquired a new HR infection [27]. An HPV test that can distinguish clinically relevant from transient HR-HPV infection is thus warranted. Improved specificity could be achieved with a modified approach to use of HPV DNA by increasing the threshold for test positivity, corresponding to higher HPV viral load, which is associated with persistent infection or infection further along the pathway to CIN2+ [26], and by utilising a restricted genotype approach to target a smaller number of genotypes that are most associated with cervical cancer [37]. We have shown in this study that increasing the threshold for test positivity and restricting the test to specific HR genotypes can increase the specificity of HC2 to distinguish CIN2+ by 20%. These findings are consistent with a cross-sectional study evaluating the diagnostic accuracy of GeneXpert among WLHIV in Cape Town, SA, that reported an increase in

specificity to distinguish CIN2+ from 60% using the manufacturer-defined threshold and targeting 14 HR-HPV types to 77% using a higher threshold to determine test positivity and restricting analysis to 8 HR-HPV types [31]. There is however some loss in sensitivity associated with this approach, and a balance will need to be achieved based on capacity to refer HR-HPV-positive women for colposcopy and treatment.

We also found that the specificity of HC2 varied according to ART status, with the highest specificity observed in women taking ART for more than 2 years, corresponding with lower HR-HPV and CIN2+ prevalence. These findings are consistent with that reported in a cohort of WLHIV undergoing screening in Nairobi, Kenya [5], and Johannesburg, SA [4]. In the future, all women newly diagnosed with HIV should start ART immediately [38], irrespective of CD4+ cell count. It is expected that women starting ART at the time of HIV diagnosis who experience a shorter duration of immunosuppression, or none, will have lower risk of HR-HPV persistence, CIN2/3 incidence, and cervical cancer compared to WLHIV who may have initiated ART according to older guidelines [10]. As a consequence, the specificity of HPV-DNA-based approaches may be higher in these women due to the lower prevalence of transient or non-clinically relevant HPV infections. An HPV-based strategy using a modified threshold, with or without a restricted genotype approach, could be a highly accurate and reproducible screening strategy in these women. However, there will remain a significant proportion of WLHIV who started ART under older guidelines and at lower CD4+ cell count, or women in settings where early access to ART may be a challenge, who remain at elevated risk. HPV-based test specificity remained low in these women in our study, irrespective of the threshold for test positivity or use of a restricted genotype approach. In the short term, it may be necessary to consider a risk stratification approach with alternative screening strategies for women with poorly controlled HIV, and if HPV-based tests are used for screening, this group may require a second test in triage or repeat testing over time due to the low specificity of a one-time HPV test among these women.

Alternative approaches to the use of HPV tests could include repeat HPV DNA testing over time, which may distinguish HR-HPV persistent infection associated with CIN2+ from transient infections. We found in this study that 74% of WLHIV with CIN2+ detected at endline had type-specific persistence from baseline, compared with 23% of women without CIN2+ at endline. While such an approach may result in fewer women being unnecessarily treated or referred to colposcopy, the limitation is the potential for loss to follow-up of screen-positive women compared to a one-time HPV DNA test. On the other hand, repeat testing over a shorter interval (e.g., 6 months) may be a feasible approach to integrate in routine HIV care, where WLHIV may be more frequently followed. Further data collection on the effectiveness and feasibility of such an approach is warranted.

VIA is commonly used in LMICs, but we have shown it has low sensitivity for CIN2 lesions in WLHIV, consistent with other studies in Africa [4,5,36], but has higher sensitivity for CIN3 + in BF only. We also evaluated the diagnostic accuracy of VIA/VILI in HR-HPV-positive women, but this approach resulted in similarly low sensitivity as for VIA alone, although the addition of VILI to VIA (i.e., either test positive) improved sensitivity by approximately 15% for CIN2+/CIN3+. The combination of VIA/VILI also had better accuracy for CIN3+ compared to CIN2+ in BF, but not in SA. This may be because VIA/VILI is more frequently used as a screening test in BF compared to SA, although study nurses and midwives were trained on VIA/VILI procedures in a similar way in both settings prior to participant recruitment in this study. The difference might also be explained by the higher prevalence of other STIs and cervical inflammation among women in SA compared to women in BF [17], which could impact the visualisation of the cervix. Visual inspection methods are highly variable due to their subjective nature, and optimal performance is dependent on observer training and experience and

the availability of quality assurance, including review of digital cervicography to ensure standardisation of VIA/VILI [4,6,36,39,40], which may be challenging to implement at scale [41]. Computer-aided approaches using automated visual evaluation (AVE) could improve the accuracy and reproducibility of visual inspection methods. AVE applied to cervigrams has been evaluated in HIV-negative women in Costa Rica and shown to have higher accuracy (area under the curve [AUC] = 0.91, 95% CI 0.89–0.93) compared to conventional cytology (AUC = 0.71, 95% CI 0.65–0.77) [42] but has not yet been studied in WLHIV, although studies are ongoing.

Cytology was the strategy with the best combination of sensitivity and specificity in SA, but only when the threshold for test positivity was increased to HSIL+. Similar high accuracy of cytology for CIN2+/CIN3+ has been reported in other studies in SA, which has an established cytology-based screening programme with quality control measures routinely implemented [4]. Studies conducted in the sub-Saharan African region have reported variable sensitivity and specificity of cytology for CIN2+/CIN3+ in WLHIV [5,40,43–45]; however, in countries where established cytology services exist, strengthening cytology services should ensure high accuracy. Sensitivity of HC2 for CIN2+/CIN3+ was higher than that achieved by cytology HSIL+ in SA, but HC2 detected fewer CIN2+ and CIN3+ cases in SA compared to BF, and the reasons behind this finding are unclear. Based on genotyping using INNO-LiPA, 11% (8/76) of women with prevalent CIN2 and 8% (4/53) of women with CIN3 were HR-HPV negative; 5% of CIN2+ cases were negative for any HPV DNA. It is not uncommon to find women with CIN2+ being HR-HPV negative. A systematic review comparing the HPV type distribution in ICC biopsy and cervical cell specimens of 770 WLHIV from 21 studies in 12 African countries reported that prevalence of any HPV was 89% in biopsy samples and 95% in cervical samples [37]. Similarly, in a review of 10,575 biopsies of ICC, 85% were positive for any HPV [46], and in a sub-analysis of a large cervical cancer screening study (ATHENA), among 497 cases of CIN2+, 55 (11%) tested negative by Cobas HPV test and 12 (2.4%) were negative by all HPV tests (Cobas, Amplicor, and Linear Array) [47]. Our finding of 5% of CIN2+ cases being negative for any HPV is not dissimilar to the findings of these large international studies. It is unlikely that CIN2+ cases were misclassified, as all CIN2+ cases were verified by consensus among 5 independent pathologists [24], although the risk of misclassification cannot be eliminated.

This study has several limitations. The study maximised the chances of obtaining histological results by basing the biopsy decision on positivity of any of 3 screening tests (HC2, cytology ASCUS+, or VIA/VILI abnormal) or colposcopy (abnormal), to which all participants were subjected (96% of women underwent all tests). This approach and the threshold to trigger biopsy for histology are in excess of usual recommendations to minimise ascertainment bias. The number of post-biopsy adverse events was low; 6 (1.0%) women in BF and 4 (0.6%) in SA reported post-biopsy bleeding and/or abdominal pain. Women negative by all tests were considered to be at extremely low risk of CIN2+ since in particular HPV DNA and cytology have very high NPV for CIN2 diagnosis [48], and it is therefore unlikely that many cases would have been missed. In addition, the study built a strong review of histological results by consensus of 5 pathologists, which included all histological slides from women with a local diagnosis of CIN2+ and approximately 10% of slides from women with ≤CIN1, which showed high agreement [24]. WLHIV included in this study were recruited from 2011 to 2012, at a time when they may have started ART according to older guidelines. As such, the study population may not be representative of contemporary or future cohorts of WLHIV in the universal ART era. However, our analysis of diagnostic accuracy according to ART status and duration attempted to correct for this period effect by restricting analysis to women with controlled HIV, which corresponds to the approach recently used in a contemporary cohort of WLHIV

enrolled in 2013–2015 in the US [49]. We did not evaluate the HPV tests in the local study settings at baseline, and HC2 was conducted in France due to challenges in acquiring the careHPV assay in time for study initiation. However, careHPV testing was conducted locally at study sites at endline and showed equivalent diagnostic accuracy as HC2 in a head-to-head comparison, previously published [21].

## Conclusion

HPV-based tests may be sufficient as a screening strategy in WLHIV if a restricted genotype approach is utilised and a higher threshold for test positivity is established. Molecular-based tests such as HPV tests have the added advantage of being automatable and less prone to training and interpretational errors than morphological tests such as VIA/VILI and cytology and can be performed using the same clinician-collected or self-collected sample, thereby simplifying sample collection, which may facilitate cervical cancer screening without the need for women to attend clinical services. Cytology remains optimal in settings with an existing cytology-based programme, such as SA. ART users with low or unknown nadir CD4+ cell count and ART-naïve women should be screened frequently, although the optimal screening intervals remain unclear. Although cervical cancer screening is not widely implemented in LMICs, integration of cervical cancer screening within HIV treatment services would ensure that women at high risk of developing cervical cancer precursor lesions are screened, and would lead to continuity in primary prevention, favouring early detection and management of HPV-related cervical lesions with minimal loss to follow-up [50]. More longitudinal data are needed on the effectiveness and cost-effectiveness of different cervical cancer screening strategies in cervical cancer reduction in WLHIV.

## Supporting information

**S1 Checklist. STARD checklist.**
(DOCX)

**S1 Table. Diagnostic accuracy of screening strategies for detection of prevalent CIN3 + among 554 unscreened WLHIV in BF.**
(DOCX)

**S2 Table. Diagnostic accuracy of screening strategies for detection of prevalent CIN3 + among 576 unscreened WLHIV in SA.**
(DOCX)

**S3 Table. Diagnostic accuracy of screening strategies for detection of prevalent CIN2 + among 554 unscreened WLHIV in BF.**
(DOCX)

**S4 Table. Diagnostic accuracy of screening strategies for detection of prevalent CIN2 + among 576 unscreened WLHIV in SA.**
(DOCX)

**S5 Table. Diagnostic accuracy of screening tests (HPV DNA, VIA, and cytology) for CIN2 +/CIN3+, by age group.**
(DOCX)

**S6 Table. Diagnostic accuracy of cervical cancer screening strategies for CIN3+ detection among women living with HIV (WLHIV), stratified by ART status.**
(DOCX)

**S7 Table. Diagnostic accuracy of cervical cancer screening strategies for CIN2+ detection among women living with HIV (WLHIV), stratified by ART status.**
(DOCX)

**S8 Table. HPV test positivity at baseline and endline and type-specific HR-HPV infection according to CIN status at baseline and endline among 933 WLHIV followed over a median 16 months in Burkina Faso and South Africa.**
(DOCX)

**S9 Table. CIN2+ incidence at endline among baseline screen-negative women, by ART status (countries combined).**
(DOCX)

**S10 Table. Diagnostic accuracy of endline screening strategies for detection of cumulative CIN2+ prevalence among 431 WLHIV in BF and 415 in SA, excluding women who were treated for baseline CIN2+.**
(DOCX)

**S1 Text. HARP study plan of analysis.**
(PDF)

## Acknowledgments

careHPV test kits and testing systems were obtained through a Qiagen donation programme.

Contributing members of the HARP Study Group included the following: A. Chikandiwa, E. Cutler, S. Delany-Moretlwe, D. A. Lewis, M. P. Magooa, V. Maseko, P. Michelow, B. Muzah, T. Omar, and A. Puren (Johannesburg, SA); F. Djigma, J. Drabo, O. Goumbri-Lompo, N. Meda, B. Sawadogo, J. Simporé, A. Yonli, and S. Zan (Ouagadougou, BF); V. Costes, M. N. Didelot, S. Doutre, N. Leventoux, N. Nagot, J. Ngou, and M. Segondy (Montpellier, France); and A. Devine, C. Gilham, L. Gibson, H. Kelly, R. Legood, P. Mayaud, and H. A. Weiss (London, UK).

The HARP Study Group also wishes to thank its international scientific advisory group, consisting of Prof. C. Lacey (Chair, University of York, UK), Prof. Y. Qiao (Chinese Academy of Medical Sciences and Peking Union Medical College, Beijing, China), Prof. M. Chirenje (University of Harare, Zimbabwe), and Prof. S. de Sanjosé (Institut Català d'Oncologia, Barcelona, Spain), and the members of the HARP Endpoint Committee, including T. Omar (Johannesburg, SA); O. Goumbri-Lompo (Ouagadougou, BF); S. Doutre, N. Leventoux, and V. Costes (Montpellier, France); and O. Clavero (Institut Català d'Oncologia, Barcelona, Spain).

## Author Contributions

**Conceptualization:** Michel Segondy, Nicolas Nagot, Nicolas Meda, Sinead Delany-Moretlwe, Philippe Mayaud.

**Data curation:** Helen A. Kelly, Clare Gilham, Michel Segondy, Philippe Mayaud.

**Formal analysis:** Helen A. Kelly, Clare Gilham, Michel Segondy.

**Funding acquisition:** Philippe Mayaud.

**Investigation:** Helen A. Kelly, Admire Chikandiwa, Bernard Sawadogo, Clare Gilham, Pamela Michelow, Olga Goumbri Lompo, Tanvier Omar, Souleymane Zan, Precious Magooa, Michel Segondy, Nicolas Nagot, Nicolas Meda, Sinead Delany-Moretlwe, Philippe Mayaud.

**Methodology:** Clare Gilham, Pamela Michelow, Olga Goumbri Lompo, Tanvier Omar, Souleymane Zan, Precious Magooa, Michel Segondy, Nicolas Nagot, Nicolas Meda, Sinead Delany-Moretlwe, Philippe Mayaud.

**Project administration:** Helen A. Kelly, Admire Chikandiwa, Bernard Sawadogo, Nicolas Meda, Sinead Delany-Moretlwe, Philippe Mayaud.

**Resources:** Helen A. Kelly, Admire Chikandiwa, Bernard Sawadogo, Clare Gilham, Pamela Michelow, Olga Goumbri Lompo, Tanvier Omar, Souleymane Zan, Precious Magooa, Michel Segondy, Nicolas Meda, Sinead Delany-Moretlwe, Philippe Mayaud.

**Software:** Clare Gilham.

**Supervision:** Helen A. Kelly, Admire Chikandiwa, Bernard Sawadogo, Michel Segondy, Nicolas Nagot, Nicolas Meda, Sinead Delany-Moretlwe, Philippe Mayaud.

**Validation:** Clare Gilham, Pamela Michelow, Olga Goumbri Lompo, Tanvier Omar, Souleymane Zan, Precious Magooa, Michel Segondy, Sinead Delany-Moretlwe, Philippe Mayaud.

**Visualization:** Pamela Michelow, Olga Goumbri Lompo, Tanvier Omar, Souleymane Zan, Philippe Mayaud.

**Writing – original draft:** Helen A. Kelly, Clare Gilham, Sinead Delany-Moretlwe, Philippe Mayaud.

**Writing – review & editing:** Helen A. Kelly, Admire Chikandiwa, Bernard Sawadogo, Clare Gilham, Pamela Michelow, Olga Goumbri Lompo, Tanvier Omar, Souleymane Zan, Precious Magooa, Michel Segondy, Nicolas Nagot, Nicolas Meda, Sinead Delany-Moretlwe, Philippe Mayaud.

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
