## [Editor Report · Decision Letter 0]

1 Sep 2020

Dear Dr Kelly, 

Thank you for submitting your manuscript entitled "Defining optimal cervical cancer screen and triage strategies for women living with HIV-1 in Sub-Saharan Africa: a diagnostic accuracy study" for consideration by PLOS Medicine.

Your manuscript has now been evaluated by the PLOS Medicine editorial staff and I am writing to let you know that we would like to send your submission out for external peer review.

Kind regards,

Helen Howard, for Clare Stone PhD 

Acting Editor-in-Chief

PLOS Medicine 

plosmedicine.org

---

## [Decision Letter · Decision Letter 1]

30 Sep 2020

Dear Dr. Kelly,

Thank you very much for submitting your manuscript "Defining optimal cervical cancer screen and triage strategies for women living with HIV-1 in Sub-Saharan Africa: a diagnostic accuracy study" (PMEDICINE-D-20-04079R1) for consideration at PLOS Medicine. 

Your paper was evaluated by a senior editor and discussed among all the editors here. It was also evaluated by three independent reviewers, including a statistical reviewer. The reviews are appended at the bottom of this email and any accompanying reviewer attachments can be seen via the link below:

[LINK]

In light of these reviews, I am afraid that we will not be able to accept the manuscript for publication in the journal in its current form, but we would like to consider a revised version that addresses the reviewers' and editors' comments. Obviously we cannot make any decision about publication until we have seen the revised manuscript and your response, and we plan to seek re-review by one or more of the reviewers. 

We expect to receive your revised manuscript by Oct 21 2020 11:59PM. Please email us (plosmedicine@plos.org) if you have any questions or concerns.

We look forward to receiving your revised manuscript. 

Sincerely,

Emma Veitch, PhD

PLOS Medicine

On behalf of Artur Arikainen, PhD, Associate Editor, 

PLOS Medicine

plosmedicine.org

Requests from the editors (writeup/formatting changes for the journal):

*Please structure your abstract using the PLOS Medicine headings (Background, Methods and Findings, Conclusions) - "methods and findings" is a single subsection. In the last sentence of the Abstract Methods and Findings section, please include a brief note about any key limitation(s) of the study's methodology.

*It would also be good to include in the Abstract section some information about the dates of enrollment into the two cohorts involved in this study. If length is an issue for the abstract (with more methods info and limitations now mentioned), some of the less informative results could be omitted. 

*At this stage, we ask that you include a short, non-technical Author Summary of your research to make findings accessible to a wide audience that includes both scientists and non-scientists. The Author Summary should immediately follow the Abstract in your revised manuscript. This text is subject to editorial change and should be distinct from the scientific abstract. Please see our author guidelines for more information: https://journals.plos.org/plosmedicine/s/revising-your-manuscript#loc-author-summary

*It would be good to clarify in the paper whether the analytical approach reported here corresponded to one laid out in a prospective protocol or analysis plan. Please state this (either way) early in the Methods section.

*The paper is clear that the STARD guideline was used in reporting (and the checklist is appended as a supporting information file) - however we'd recommend that the authors call out the SI file corresponding to the checklist in the Methods section where this is mentioned. In addition, some of the STARD items are blank, and some of these correspond to issues noted by one reviewer as being unclear in the study, so it would be good to clarify this. Finally when completing the checklist for resubmission, please use section and paragraph numbers, rather than page numbers (because the latter are amended when the paper is reformatted for publication). 

Comments from the reviewers:

Reviewer #1: This is an interesting diagnostic accuracy study on cervical cancer screening and triage strategies for women living with HIV-1 in Sub-Saharan Africa. However, there are quite a few issues needing attention.

1) The title of "Defining optimal cervical cancer screen and triage strategies..." is a bit misleading. This is basically an observational studies on the diagnostic accuracies of a range of cervical cancer tests on the two cohorts for BF and SA. To develop and validate a diagnostics test, needs to develop/establish a test in a training set and then validate the test in the independent/external validation set, the performance of the diagnostic test needs to be evaluated for both discrimination and calibration. So far, we didn't see any of the above in the paper, therefore it's mainly an observational study reporting on the test results of a range of diagnostic tests.

2) As this is a diagnostic study, it's not clear what is the gold standard of all the tests. Also, it could be better structured and written in a way - primary outcome, secondary outcome, incremental test accuracies of test 1, 2 and 3. Now it's all over the place. What exactly is the research question? What's the rationale and link of involving two cohorts from BF and SA? It's like two separate studies. Also, what's the point of baseline and endline measurements? For the diagnostic studies, it is 'on the spot' test which is very different from prognostic test with follow-ups. Basically the follow-ups in the study didn't contribute to the evaluation of the tests. It would be good for authors to refer to the TRIPOD statement to the principles of diagnostic and prognostic studies.

3) ROC curve is very important for the reporting of diagnostic tests. However, it's not mentioned at all in the statistics analysis section in methods. In Figure 2 and 3, there needs to be curves there not just the dots, and ROC values need to have 95% CIs. Also, the ROC values with 95% CI need to be presented in all the tables for test performance.

4) In statistics analysis of the methods section, it says "We modelled the diagnostic accuracy of restricted genotyping approach using results of INNO-LiPA genotyping assay in the following combinations...", however, it's not clear what exactly the statistical model being used for this?

5) Table 1. P-value is missing for the variable "HIV-1 PVL in ART users".

Reviewer #2: Thank you for this manuscript. This is a very important topic and a well written manuscript. In this time when the need for a specific cervical pre-cancer/cancer screening and prevention strategy for women living with HIV, such a research and manuscript adds to the accumulating evidence to inform policy making. I have very few comments and suggestions to help improve the manuscript:

1. line 9: I suggest you make "cervical screening" to read cervical cancer screening. This will be clearer to the readers and also consistent with the rest of the manuscript

2. Line 64-54: rephrase possible to read 'All participants were referred for colposcopy performed AT A median OF 12 weeks (interquartile range [IQR]: 8-15)...'

3. tables 2-5: I understand you have tried to put all the required information on the tables but as they look now, they are difficult to read. Kindly see what can be done to improve the legibility. I think they are very important tables and so any efforts to make them more legible is important.

4. line 425: it should read '.....test positivity IS established.....'

Reviewer #3: The authors tackle a key unanswered question for women living with HIV (WLHIV) globally. Cervical cancer has become the leading cause of death for WLHIV with access to ART in Africa, but achievable strategies to mitigate this burden have been illusive due to challenges of scale, prevalence of dysplasia, and pathology/cytology/PCR capacity. In this project, they prospectively enroll WLHIV in Burkina Faso and South Africa (illustrative of the diversity of HIV/HPV prevalence and diagnostic capacities present where bulk of WLHIV live) to rigorously assess performance characteristics of visual, cytologic, and molecular approaches to screening. The core finding that molecular testing is superior (or equivalent to high performing cytology program) is a key piece of knowledge to move field/programs forward. However, as currently presented these findings are challenging for a general audience to gasps and understand importance. Revision for improved readability and focus could greatly improve impact.

Suggestions:

Major.

1) Improve readability of abstract/manuscript for general medical audience- Abstract reads as though two different studies (BF and RSA), but could be improved by noting that evaluating in low and high/middle-income contexts with differing HIV and HPV prevalences for improved generalizability. Suggest considering merging findings as seems like molecular testing optimal strategy in both settings (marginal difference with cytology in RSA), for detecting cancers and reducing specialist-dependant services.

2) More directly assess impact of weaknesses on findings/confidence. The key weaknesses include old study (collected 2011/12, still reflective of current environment?), unblinding of pathologists (how much bias would need to be present to modify findings?), use of careHPV (challenging technology to use, and endpoints seem to only be HC-II done in Europe, reflective of reality in BF/RSA implementation?), does different geography of HPV prevalence affect the conclusions, and can any intervention be effective if takes 8-14mo to treat dysplasia and 37% got no treatment?

3) Age triage. Many programs considering threshold of 30yrs to improve specificity for HPV testing. Does this subgroup have different performance characteristics>

Minor

1) Unsure why some patients got VIA or VILI or both

2) For an HIV audience, they would expect viral suppression as a categorical variable rather than continuous. And proportion with viral failure on ART (often >1000 copies)

3) Introduction and elsewhere suggest relationship between HPV vaccination and HPVtesting, but not related

4) Discussion long and lots of extrapolations re: CD4, ART timing beyond the data of this project. suggest focusing message.

[LINK]

---

## [Decision Letter · Decision Letter 2]

9 Dec 2020

Dear Dr. Kelly,

Thank you very much for re-submitting your manuscript "Diagnostic accuracy of cervical cancer screening and screening-triage strategies among women living with HIV-1 in Africa" (PMEDICINE-D-20-04079R2) for review by PLOS Medicine.

I have discussed the paper with my colleagues and the academic editor and it was also seen again by three reviewers. I am pleased to say that provided the remaining editorial and production issues are dealt with we are planning to accept the paper for publication in the journal.

[LINK]

We look forward to receiving the revised manuscript by Dec 16 2020 11:59PM.   

Sincerely,

Artur Arikainen, 

Associate Editor 

PLOS Medicine

plosmedicine.org

Requests from Editors:

1. Title: Please revise to: “Diagnostic accuracy of cervical cancer screening and screening-triage strategies among women living with HIV-1 in Burkina Faso and South Africa: A cohort study”

2. Please provide the data underlying your study, and update the Data Availability Statement accordingly. We will need to check this prior to acceptance.

3. Please include line numbers in the margin throughout, including the abstract and author summary.

4. Throughout, please use the following capitalisation: "sub-Saharan”.

5. Abstract:

a. Please give more summary demographic information, eg. median age at recruitment, prior screenings.

b. Please include p values for comparisons.

c. Please include 95% CIs for these results: “CIN2+ incidence over 16 months was highest among VIA baseline screen-negative women (2.2%) and HC-II triage with VIA (2.1%) and lowest among HC-II screen-negative women (0.5%).”

d. We had some difficulty locating the data in these statements for verification; please indicate which Table these can be found in:

i. “Triage of HC-II positive women with VIA/VILI reduced the number of colposcopy referrals, but with loss in sensitivity for CIN2+ (58.1%) but not for CIN3+: (84.6%).”

ii. “HC-II had higher sensitivity (86.8%, 95%CI: 79.7-92.1) but low specificity (48.2%, 95%CI; 43.5-53.0) resulting in almost twice as many referrals.”

iii. “Compared to HC-II, triage of HC-II positive women with HSIL+ resulted in 40% reduction in colposcopy referrals but was associated with some loss in sensitivity.”

e. Conclusions: Please start with “In this cohort study, …”

f. Conclusions: Please mention WLHIV as well.

6. Author summary: 

a. Please add “(WLHIV)” after first use.

b. Please correct: “precancerous lesions cervical” to “precancerous cervical lesions” 

c. Please define/clarify “cytology” and “colposcopy” to a lay reader.

d. Please avoid the phrasing “false-positive”, and instead merely describe this subset of patients, eg …infections that have not progressed to a cytology-detectable lesion…“”. Please also clarify the “second test” as being also for HPV.

e. Please define “ART” at first use.

f. Please delete “and in COVID-19 era”

7. Please make your reference citations not superscript, and include a space between the text and the brackets, eg: “…increase screening coverage and uptake [1].” Please remove spaces from within citations themselves.

8. Line 54: Please include days in the recruitment dates.

9. Line 90: Please remove registered trademark symbol.

10. PLOS does not permit "data not shown” (eg. lines 176, 264, 293). Please remove this claim, or do one of the following:

a) If you are the owner of the data relevant to this claim, please provide the data in accordance with the PLOS data policy, and update your Data Availability Statement as needed.

b) If the data not shown refer to a study from another group that has not been published, please cite personal communication in your manuscript text (it should not be included in the reference section). Please provide the name of the individual, the affiliation, and date of communication. The individual must provide PLOS Medicine written permission to be named for this purpose.

c) For any other circumstance, please contact the journal office ASAP.

11. Table 1 and throughout: Please show exact p values over 0.001, or p<0.001 otherwise.

12. Line 346: Typo “..”

13. Page 27: Please remove the declaration of interests and author contributions – these are taken from the online submission form.

14. Please double check and correct the URLs for references 1, 2, 18. 

15. Please provide a URL or DOI for references 11 and 33.

16. Please remove duplicate URLs from references 14 and 15.

17. Please include a title for reference 22.

18. Please list the short journal name for “PLoS ONE” consistently in references.

19. Please rename the STARD checklist file as “S1 Checklist”. Line 153: Please update the reference to this file.

20. Please rename the analysis plan file as “S1 Text”. Please cite this file in the Methods, as with the STARD checklist.

---

Comments from Reviewers:

Reviewer #1: Many thanks authors for their tremendous effort to improve the manuscript. All my comments were comprehensively addressed. I am satisfied with the response and revision. No further issues needing attention. 

Reviewer #2: Thank you for the extensive revision. I am satisfied with how the issues raised have been addressed. But I will suggest a final read through of the clean version for editing purpose. 

Reviewer #3: The presented work is important and is substantially improved following peer review. It will be of high utility to the field.

[LINK]

---

## [Editor Report · Decision Letter 3]

22 Dec 2020

Dear Dr Kelly, 

On behalf of my colleagues and the guest Academic Editor, Scott L. Dryden-Peterson, I am pleased to inform you that we have agreed to publish your manuscript "Diagnostic accuracy of cervical cancer screening and screening-triage strategies among women living with HIV-1 in Burkina Faso and South Africa: A cohort study" (PMEDICINE-D-20-04079R3) in PLOS Medicine.

PRESS

Sincerely, 

Artur A. Arikainen 

Associate Editor 

PLOS Medicine